# IGF2 Mediates Resistance to Isoform-Selective-Inhibitors of the PI3K in HPV Positive Head and Neck Cancer

**DOI:** 10.3390/cancers13092250

**Published:** 2021-05-07

**Authors:** Mai Badarni, Manu Prasad, Artemiy Golden, Baisali Bhattacharya, Liron Levin, Ksenia M. Yegodayev, Orr Dimitstein, Ben-Zion Joshua, Limor Cohen, Ekaterina Khrameeva, Dexin Kong, Angel Porgador, Alex Braiman, Jennifer R. Grandis, Barak Rotblat, Moshe Elkabets

**Affiliations:** 1The Shraga Segal Department of Microbiology, Immunology and Genetics, Faculty of Health Science, Ben-Gurion University of the Negev, Beer-Sheva 84105, Israel; maibda@post.bgu.ac.il (M.B.); manupras@post.bgu.ac.il (M.P.); baisali@post.bgu.ac.il (B.B.); Kseniay@post.bgu.ac.il (K.M.Y.); ayashli@post.bgu.ac.il (L.C.); angel@bgu.ac.il (A.P.); braiman@bgu.ac.il (A.B.); 2Faculty of Health Sciences, Ben-Gurion University of the Negev, Beer-Sheva 84105, Israel; orrdi@clalit.org.il (O.D.); benzionj@bmc.gov.il (B.-Z.J.); 3Center of Life Sciences, Skolkovo Institute of Science and Technology, 121205 Moscow, Russia; golden.artemiy@gmail.com (A.G.); E.Khrameeva@skoltech.ru (E.K.); 4Bioinformatics Core Facility, National Institute for Biotechnology in the Negev, Ben-Gurion University of the Negev, Beer-Sheva 84105, Israel; levinl@post.bgu.ac.il; 5The National Institute for Biotechnology in the Negev, Ben-Gurion University of the Negev, Beer Sheva 84105, Israel; 6Department of Otolaryngology—Head and Neck Surgery, Soroka University Medical Center, Beer-Sheva 84105, Israel; 7Department of Otorhinolaryngology and Head & Neck Surgery, Barzilay Medical Center, Ashkelon 7830604, Israel; 8School of Pharmaceutical Sciences, Tianjin Medical University, Tianjin 300070, China; kongdexin@tmu.edu.cn; 9Department of Otolaryngology—Head and Neck Surgery, University of California San Francisco, San Francisco, CA 94143, USA; Jennifer.Grandis@ucsf.edu; 10Department of Life Sciences, Faculty of Life Science, Ben-Gurion University of the Negev, Beer-Sheva 84105, Israel

**Keywords:** head and neck cancer, HPV, IGF2, IGF1R, PI3K, therapy resistance, drug combinations

## Abstract

**Simple Summary:**

In the current study, we delineate the molecular mechanisms of acquisition of resistance to two isoform-selective inhibitors of PI3K (isiPI3K), alpelisib and taselisib, in human papillomavirus positive head and neck cell lines. By comparing RNA sequencing of isiPI3K-sensitive tumor cells and their corresponding isiPI3K-acquired-resistant tumor cells, we found that overexpression of insulin growth factor 2 (IGF2) is associated with the resistance phenotype. We further demonstrated by gain and loss of function studies that IGF2 plays a causative role in limiting the sensitivity of human papillomavirus-positive head and neck cell lines. Moreover, we show that blocking IGF2 stimulation activity, using an inhibitor of the IGF1 receptor (IGF1R), enhances isiPI3K efficacy and displays a synergistic anti-tumor effect in vitro and superior anti-tumor activity ex vivo and in vivo.

**Abstract:**

Over 50% of human papilloma positive head-and-neck cancer (HNC^HPV+^) patients harbor genomic-alterations in *PIK3CA*, leading to hyperactivation of the phosphatidylinositol-4, 5-bisphosphate 3-kinase (PI3K) pathway. Nevertheless, despite PI3K pathway activation in HNC^HPV+^ tumors, the anti-tumor activities of PI3K pathway inhibitors are moderate, mostly due to the emergence of resistance. Thus, for potent and long-term tumor management, drugs blocking resistance mechanisms should be combined with PI3K inhibitors. Here, we delineate the molecular mechanisms of the acquisition of resistance to two isoform-selective inhibitors of PI3K (isiPI3K), alpelisib (BYL719) and taselisib (GDC0032), in HNC^HPV+^ cell lines. By comparing the transcriptional landscape of isiPI3K-sensitive tumor cells with that of their corresponding isiPI3K-acquired-resistant tumor cells, we found upregulation of insulin growth factor 2 (IGF2) in the resistant cells. Mechanistically, we show that upon isiPI3K treatment, isiPI3K-sensitive tumor cells upregulate the expression of IGF2 to induce cell proliferation via the activation of the IGF1 receptor (IGF1R). Stimulating tumor cells with recombinant IGF2 limited isiPI3K efficacy and released treated cells from S phase arrest. Knocking-down IGF2 with siRNA, or blocking IGF1R with AEW541, resulted in superior anti-tumor activity of isiPI3K in vitro and ex vivo. In vivo, the combination of isiPI3K and IGF1R inhibitor induced stable disease in mice bearing either tumors generated by the HNC^HPV+^ UM-SCC47 cell line or HPV+ patient-derived xenografts. These findings indicate that IGF2 and the IGF2/IGF1R pathway may constitute new targets for combination therapies to enhance the efficacy of PI3K inhibitors for the treatment of HNC^HPV+^.

## 1. Introduction

Head and neck cancer (HNC) is an aggressive cancer with a five-year survival rate of ~60% [1]. Every year 890,000 new patients are diagnosed with HNC and 450,000 patients die from this malignancy worldwide [2]. Alcohol consumption, smoking, and infection with human papillomaviruses (HPV) are the major causes of HNC (reviewed in Reference [3].) Over the past few decades, the incidence of HPV negative HNC (HNC^HPV−^) has slowly been declining, while the numbers of patients with HPV positive HNC (HNC^HPV+^) have been increasing dramatically [4,5,6]. The response of HNC^HPV+^ patients to the standard treatment regimen is generally good, with these treatments reducing the risk of death by 60% [7]. Nevertheless, a subpopulation of these patients does not respond to FDA-approved therapies and succumb to recurrent or metastatic disease [7,8]. Accordingly, there is an urgent medical need to develop more effective treatments for patients with HNC^HPV+^.

HPV are small, double-stranded DNA viruses that infect the basal layer of squamous epithelia [9,10]. Over 200 types of HPV are known; these are divided into low-risk and high-risk groups based on their ability to drive malignancy [11]. High-risk HPV^16^ is the major type in HNC^HPV+^ patients [12,13]. At the molecular level, HNC^HPV^^+^ differs from HNC^HPV−^ in that it expresses the E6 and E7 oncoproteins, which increase cell proliferation and genetic instability, mainly by blocking the function of the crucial cell cycle regulators p53 and Rb, respectively [14,15,16]. It is known that HNC^HPV+^ tumors express p16 due to the negative feedback induced by E7; therefore, assessing p16 expression by immunohistochemistry (IHC) is the most commonly applied test in the clinic to assess the HPV status of the tumor [17].

Comprehensive genomic analysis of HNC^HPV+^ tumor data, obtained from the Cancer Genome Atlas (TCGA), showed a low frequency of genomic alteration in *TP53* (p53; 3%) and no mutations in *CDKN2A* (p16), when compared to HNC^HPV−^, in which 84% of the cases exhibit mutations in p53 and 57% cases exhibit mutations in p16, respectively [18]. In contrast, genomic alterations in the *PIK3CA* gene are observed in 56% of HPV^+^ cases and in 34% of HPV− tumors. In HNC^HPV+^, the *PIK3CA* gene, which encodes the p110a isoform of the phosphoinositide 3-kinase (PI3K) complex [18], is either mutated in the region encoding the protein’s helical or kinase domains, or amplified, along with the other genes in the 3q26 locus (*PIK3CA* gene located in 3q26.32). These genomic alterations (point mutations or amplification) in *PIK3CA* result in hyperactivation of the PI3K pathway, leading to cellular transformation and to enhanced tumor cell growth, survival, and motility, all of which contribute to tumor progression (reviewed in Reference [19]). Tumors with genomic alterations in the *PIK3CA* gene are often more susceptible to treatment with isoform-selective inhibitors of PI3K (isiPI3K) than wild-type (WT) *PIK3CA* tumors [20,21,22,23,24,25]. Nevertheless, the *PIK3CA* status is not a surrogate marker of response in HNC^HPV+^ [26,27,28,29], underscoring the need to study the mechanisms of resistance and response to isiPI3K in *PIK3CA* WT HNC^HPV+^.

Alpelisib (BYL719) is an FDA-approved isiPI3K [30] that binds the p110 alpha subunit of PI3K and is inducing tumor growth arrest in *PIK3CA* altered solid cancers, including in HNC^HPV−^ [23,29,31,32,33]. A similar anti-tumor response to a different isiPI3K, taselisib (GDC0032, a beta-sparing PI3K inhibitor), was observed in phase 1b clinical trials [34,35]. In a phase I basket trial, ~15% of *PIK3CA*-mutated HNC patients responded to isiPI3Ks, independently of their HPV status. Despite the intensive efforts that have been invested in the development of isiPI3Ks, the clinical benefit of those agents remains limited, as all patients experienced tumor relapse due to the acquisition of resistance. These clinical findings highlight the need to reveal the mechanisms of resistance to isiPI3K so as to facilitate the development of novel rationale-based therapies for HNC^HPV+^ patients. Here, we describe activation of IGF1R signaling as a mechanism of resistance to two isiPI3Ks, GDC0032 and BYL719, in HNC^HPV+^, where co-targeting prevents resistance and improves isiPI3K efficacy in HNC^HPV+^.

## 2. Materials and Methods

### 2.1. Tumor Cell Lines

HNC^HPV+^ cell lines were kindly provided by Reidar Grénman, Department of Otorhinolaryngology—Head & Neck Surgery, Turku University, Finland. The cells were maintained at 37 °C in a humidified atmosphere and 5% CO_2_. The cells were cultured in 10% fetal bovine serum (FBS) Dulbecco’s Modified Eagle’s medium (DMEM), supplemented with 1% l-glutamine 200 mM and 100 units each of penicillin and streptomycin. Acquired resistance cell lines were grown in presence of BYL719 (2 μM) or GDC0032 (500 nM), which was added to the medium every 3 days. Cells were routinely tested for mycoplasma infection and treated with appropriate antibiotics as needed (De-Plasma, TOKU-E, D022).

### 2.2. IC_50_ and Synergy Assay

Cells were seeded in 96-well plates, treated with increasing concentrations of the drugs being tested (0–10 μM BYL719 or 0–2 μM GDC0032 and 0–10 μM AEW451) and allowed to proliferate for 4 days. Cells were then fixed and stained with crystal violet (1 g/L) for 30 min at room temperature. Following rinsing, crystal violet was dissolved with 10% acetic acid, and absorbance was measured at 570 nm (BioTek Epoch spectrophotometer, Winooski, Vermont, USA). IC50 values were calculated using the GraphPad Software. For the synergy assays, the proliferation of the cells in the different treatment groups was determined as the percentage of control (DMSO-treated) cells, and the percent of growth inhibition was calculated. ZIP Synergy scores were calculated using the Synergyfinder website.

### 2.3. Cell Proliferation Assay

Cells were seeded in 24-well plates, treated with drugs being tested, and allowed to proliferate for 4 days. Cells were then fixed and stained with crystal violet (1 g/L) for 30 min at room temperature. Following rinsing, crystal violet was dissolved with 10% acetic acid, and absorbance was measured at 570 nm (BioTek Epoch spectrophotometer). Results were determined as a percentage of the control (DMSO-treated) cells. For live cell imaging (Figure 1D), the real-time cell history recorder JULI Stage was used to record cell confluence every 6 h.

### 2.4. Cell Cycle

Cells were seeded in 6-well plates and treated with the drugs being tested for 3 days. Cells, together with the supernatant, were collected into 15-mL tubes and centrifuged for 10 min at 4 °C. The pellet was fixed at −20 °C with 70% ethanol and stored for at least 24 h at −20 °C. Thereafter, the pellet was washed twice with ice-cold 1 × PBS, treated with 100 μL of RNase solution (100 μg/μL) for 30 min at 37 °C, and stained in the dark with 200 μL of propidium iodide solution (100 μg/mL) for 30 min at room temperature. The cell cycle phase was analyzed using FlowJo software v8.8.7.

### 2.5. RNAseq

RNA was extracted from sensitive and isiPI3K-acquired resistance UM-SCC47 and UT-SCC60A cell lines and sequenced. Three replicates of each of the sensitive and resistant cell lines were cultured in drug-free medium for 48 h, after which RNA was extracted using the RNeasy mini kit (74104, Qiagen, Hilden, Germany) according to the manufacturer’s instructions. RNA-seq libraries were prepared as described previously [36] Sequencing was performed with a Nextseq5000 sequencer using all four lanes. Analysis of the raw sequence reads was carried out using the NeatSeq-Flow platform [37]. The sequences were quality trimmed and filtered using Trim Galore (V0.4.5). Alignment of the reads to the human genome (version GRCh38.91) was done with STAR [38]. The number of reads per gene per sample was counted using RSEM [39]. Quality assessment (QA) of the process was carried out using FASTQC and MultiQC [40]. Gene annotation was done based on the human genome assembly downloaded from Ensembl/BioMart. Statistical analysis for identification of differentially expressed genes was performed using the DESeq2 R package via the NeatSeq-Flow platform [37].

### 2.6. Real-Time PCR

Total RNA was isolated from the cultured cells using an ISOLATE II RNA Mini Kit (Bioline, Meridian Bioscience, London, England) according to the manufacturer’s instructions; 1 μg RNA was converted into cDNA using qScript™ cDNA synthesis kit (95047-100, Quanta bioscience, Beverly, MA, USA) according to the manufacturer’s instructions. Real-time PCR was performed using Primetime Gene Expression Master Mix (1055770, IDT, Coralville, IA, USA), with matching Taqman probes purchased from IDT, in Roche light cycler 480 II. Analysis was performed with LightCycler 480 Software, Version 1.5.1. Fold change was calculated using the ΔΔCt method. Results were normalized to GAPDH levels and presented as a fold change relative to the control cells.

### 2.7. siRNA Transfection

For transient silencing of IGF2, cells were seeded in 24-well plates for 24 h and then transfected using Lipofectamine^®^ 3000 Reagent (L3000015, Invitrogen, CA, USA) according to the manufacturer’s protocol, with an siRNA non-targeting control sequence (IDT, 51-01-14-04) and a IGF2 gene targeting sequence (hs.Ri.IGF2.13.1, IDT, Coralville, IA, USA). For proliferation assay experiments, cells were treated with the relevant drugs following 24 h of transection and allowed to proliferate for an additional 3 days. 

### 2.8. Western Blotting

Cells were washed 3 times with PBS and scraped into lysis buffer (50 mM HEPES [pH 7.5], 150 mM NaCl, 1 mM EDTA, 1 mM EGTA, 10% glycerol, 1% Triton X-100, 10 μM MgCl_2_) at 4 °C. The lysis buffer was supplemented with phosphatase inhibitor cocktails (BiotoolB15001A/B) and protease inhibitor (P2714-1BTL, Sigma-Aldrich, Burlington, MA, USA). The cells in the lysis buffer were then placed on ice for 30 min, followed by 3 min of ultrasonic cell disruption. Lysates were cleared by centrifugation at 14,000 rpm for 10 min at 4 °C. Supernatants were collected, and protein concentration was calculated using Bio-Rad Protein Assay (500-0006, Hercules, CA, USA). Lysates were mixed with LDS sample buffer (4×) (B0007, Thermo Fisher Scientific, Waltham, MA, USA ) and NuPAGE Sample Reducing Agent (10×) (NP0009, Thermo Fisher Scientific, Waltham, MA, USA), and boiled at 95 °C for 5 min. Of the total lysate 20 mg were separated on NuPAGE 12.5% SDS–PAGE and blotted onto PVDF membranes (Bio-Rad Trans blot TurboTM transfer pack, 1704157, Hercules, CA, USA). Membranes were blocked for 1 h in blocking solution (5% BSA [Amresco 0332-TAM] in Tris-buffered saline (TBS) with 0.1% Tween) and then incubated with primary antibodies diluted in blocking solution, supplemented with sodium azide (S2002-5G, Sigma-Aldrich, Burlington, MA, USA). Mouse and rabbit horseradish peroxidase (HRP)-conjugated secondary antibodies were diluted in blocking solution. Protein-antibody complexes were detected by chemiluminescence (Westar Supernova (XLS3.0100, Cyanagen, Santa Clara, CA, USA) and Westar Nova 2.0 (XLS071.0250, Cyanagen, Santa Clara, CA, USA), and images were captured using the Azure C300 Chemiluminescent Western Blot Imaging System (Azure Biosystems, Dublin, CA, USA).

### 2.9. Immunohistochemistry (IHC) and Immunofluorescence (IF)

For xenograft samples, dissected tissues were fixed in a 4% paraformaldehyde (PFA) solution for 24 h at room temperature, dehydrated, and embedded in paraffin. The tissue sections were de-deparaffinized with xylene (3 times, 10 min each), and 3% H_2_O_2_ was used to block the endogenous peroxidase activity for 20 min, followed by a water wash of 5 min. For antigen retrieval, sections were then placed in citrate buffer (pH = 6) and heated in a boiling water bath (99.99 °C) for 20 min. Sections were then blocked for 1 with blocking solution (PBS, 0.1% Tween, 5% BSA), followed by incubation with a primary antibody diluted in blocking solution overnight at 4 °C. The IHC detection was performed using ABC kit (VECTASTAIN, Victor laboratories, San Francisco, CA, USA) and DAB reagent (ZYTOMED SYSTEMS, Berlin, Germany). Sections were counterstained with hematoxylin and mounted using Micromount (Surgipath, Richmond, VA, USA).

For immunofluorescence, cells were seeded on 13 mm coverslips held in 24-well plates, for 12–24 h before treatment. Cells were treated isiPI3K for 24 h. Cells were rinsed with PBS at 4 °C and fixed with 4% PFA for 30 min at room temperature. Thereafter, cells were rinsed with PBS, followed by permeabilization on ice in PBS supplemented with 0.1% Triton X-100 (MilliporeSigma, Burlington, MA, USA) for 10 min. Cells were then rinsed with PBS and blocked in blocking solution (5% BSA in Tween-PBS) for 1 h at room temperature. The cells were incubated with primary antibodies overnight at 4 °C, rinsed with PBS, and incubated with secondary antibody at room temperature for 1 h. Cells were then rinsed with PBS and mounted in DAPI Fluoromount-G (0100-20, Southern Biotech, Birmingham, AL, USA). IHC slides were digitalized using the Panoramic Scanner (3DHISTECH, (Budapest, Hungary) the equipment was purchased from Getter Bio-Med). Immunofluorescence slides we digitalized using an Olympus FV1000 confocal microscope.

### 2.10. Tumor Ex Vivo Analysis (TEVA)

TEVA was performed once the volume of the PDX (#7157) had reached ~500 mm^3^. The PDXs were excised aseptically from the mice and cut into 2 × 2 × 2 mm^3^ tissue explants. The 2 × 2 × 2 mm^3^ explants were then incubated with the different drugs for 24 h in 48-well tissue culture plates at 37 °C, 95% relative humidity and 5% CO_2_, under sterile conditions with suitable controls. The 2 × 2 × 2 mm^3^ explants incubated only in culture media without any drug served as control. DMEM (Gibco, Thermo Fisher Scientific, Waltham, MA, USA) containing 20% FBS (Gibco), 1 mM sodium pyruvate (Biological Industries, Haemek, Israel), 2 mM l-glutamine (Biological Industries), 1% penicillin–streptomycin–amphotericin (Biological Industries), 0.1 mM MEM non-essential amino acids (Biological Industries), 10 mM HEPES (Biological Industries), 1% BIO-MYC (Biological Industries), and 50 µg/mL gentamycin (Gibco) was used as the culture medium. The therapeutic agents for this part of the study were BYL719 (2 µM), GDC0032 (500 nM), and AEW541 (1 µM). KI67, TUNEL, and pMAPK staining were measured and the TEVA score was calculated using the following formula: (0.3 × KI67 staining) + (0.3 × TUNEL staining) + (0.3 × MAPK staining).

### 2.11. Establishment of Tumor Xenografts and Studies in Mice

NOD-scid IL2Rgamma^null^ (NSG) mice were purchased from Jackson Laboratory. For CDX, 6-week-old NSG mice were injected s.c. in the flank with 2 × 10^6^ cells diluted in 200 μL PBS (100 μL in each side). Tumors (60 mm^3^) developed after about 2 weeks.

PDX #7157 was kindly provided by Jennifer Grandis, Department of Otolaryngology—Head and Neck Surgery, University of California San Francisco, USA. The PDX was first implanted s.c. into the flanks of 6-week-old NSG mice for expansion. After successful expansion, the PDX was cut into equal chunks and implanted s.c. in the flanks of 6-week-old NSG mice.

For the in vivo experiments, animals were randomized into 4 groups of 6 mice per group (two tumors per mouse), and treated orally, every day, with vehicle (0.5% carboxymethylcellulose (9481-1KG, MilliporeSigma, Burlington, MA, USA), monotherapy of GDC0032 (10 mg/kg), BYL719 (25 mg/kg), or AEW541 (30 mg /kg), or a drug combination, as indicated in the text. Tumors were measured with a digital caliper twice a week, and tumor volumes were determined according to the formula: length × width^2^ × (π/6). At the end of the experiment, animals were sacrificed by subjecting them to CO_2_ inhalation, and the tumors were harvested for investigation. Tumor volumes were normalized to initial volumes and presented as an averaged percentage of the initial volumes ± SEM. Mice were housed in air-filtered laminar flow cabinets with a 12 h light/dark cycle and were fed food and water ad libitum.

### 2.12. Antibodies

Anti-pAKT Ser473 (#4060), anti-AKT (#4691), anti-pS6 S240/244 (#5364), anti-S6 (#2217), anti-pERK1/2 Thr202/Tyr204 (#9101), anti-ERK1/2 (#4695), anti-FOXO1 (#14952) and anti-FOXO3a (#2497) were purchased from Cell Signaling Technology (Danvers, MA, USA). KI67 was purchased from Vector laboratories (VP K451, San Francisco, CA, USA). Anti-Actin (0869100-CF) was purchased from MP Biomedicals (Irvine, CA, USA). Mouse, and rabbit horseradish peroxidase (HRP)-conjugated secondary antibodies were purchased from GE Healthcare (Chicago, IL, USA). Cy3-conjugated anti-rabbit secondary antibody (111-165-144) and Alexa Fluor 488–conjugated anti-mouse secondary antibody (115-545-062) were purchased from Jackson ImmunoResearch (West Baltimore, PA, USA).

### 2.13. Drugs and Reagents

Novartis Pharma AG (Basel, Switzerland) provided BYL719 and AEW541. GDC0032 was purchased from Selleckchem (Houston, TX, USA). For in vitro experiments, all drugs were dissolved in DMSO, and for in vivo administration, in 0.5% carboxymethylcellulose. Human IGF2 recombinant protein was purchased from Cell Signaling Technology (#5238) and dissolved in sterile water.

### 2.14. Statistics

Experiments were repeated 2–3 times. Statistical analysis was performed using GraphPad Prism software, and results are presented as means ± SEM. For comparisons between two groups, *p*-values were calculated using Student’s *t* test. For comparisons between multiple groups, *p*-values were calculated using one-way ANOVA. For in vivo experiments, two-way ANOVA was calculated using Tukey’s multiple comparison test. *p*-values of 0.05, 0.01, 0.001 or 0.0001 were considered statistically significant, as indicated by *, **, *** or **** in the figures.

### 2.15. Study Approval

Mice were maintained and treated according to the institutional guidelines of Ben-Gurion University of the Negev. Animal experiments were approved by the IACUC (IL.80-12-2015) and IACUC IL-29-05-2018(E). Helsinki was approved by the Soroka Medical Center (0103-17-SOR and 0421-16-SOR).

## 3. Results

### 3.1. Generation of BYL719- and GDC0032-Acquired Resistance HNC^HPV+^ Cell Lines

To study mechanisms of acquired resistance to isiPI3K in HNC^HPV+^ cell lines, we generated two acquired-resistant cell line models from UM-SCC47 and UT-SCC60A cells [41]. Specifically, we used UT-SCC60A, which is sensitive to BYL719, with half-maximal inhibitory concentration values (IC_50_) of ~2 μM, and UM-SCC47, which is sensitive to GDC0032 with IC_50_ of ~0.5 μM. To generate isiPI3K-acquired resistant cell lines, UM-SCC47 and UT-SCC60A tumor cell lines were exposed to increasing concentrations of either GDC0032 or BYL719, respectively, for four to eight months, until resistance developed. For each tumor cell line, two isogenic acquired-resistant cell lines were generated, namely, UT-SCC60A^Res1^, UT-SCC60A^Res2^, UM-SCC47^Res1^, and UM-SCC47^Res2^ (Figure 1A). The resistant cell lines exhibited increased IC_50_ values (Figure 1B and Appendix A) and showed minimal or no growth inhibition when exposed to a fixed concentration of BYL719 or GDC0032, but the sensitive cells exhibited a pronounced inhibition of cell proliferation (Figure 1C). Cell cycle analysis showed that treatment of isiPI3K-sensitive cells with isiPI3K induced S-phase arrest, while, in resistant cells, S-phase arrest was not detected (Figure 1D). 

To characterize the cellular response to isiPI3K in the context of resistance or sensitivity to isiPI3K, we treated sensitive and resistant cells with GDC0032 or BYL719 for 24 h, after which we monitored PI3K downstream signaling using western blot and anti-pS6 or pAKT antibodies (Appendix A). We found that isiPI3K treatment inhibited AKT activation in both sensitive and resistant cell lines. Nevertheless, the resistant cells alone maintained activation of S6.

### 3.2. FOXO Signaling Pathway Is Activated in isiPI3K-Resistant Cells

To identify the mechanisms underlying acquired -resistance to isiPI3K, we mapped the transcriptional landscape of isiPI3K-sensitive and isiPI3K-resistant cell lines by RNA sequencing (RNAseq). Specifically, we used the parental UM- SCC47 and UT-SCC60A and their corresponding acquired resistance models, UM- SCC47-Res2 and UT-SCC60A-Res2. Importantly, RNA extraction from the resistant cells was performed following 48 h of drug-free culture. Analysis of the RNAseq data (BH-adjusted *p*-value < 0.05 & FC > 1) identified 3176 genes that were upregulated and 3193 genes that were downregulated in resistant cells compared to the corresponding sensitive cells (Appendix A). To gain further insight into the biological functions of the genes differentially expressed in sensitive and resistant cells, we performed Kyoto Encyclopedia of Genes and Genomes (KEGG) pathway analysis. Among the five upregulated signatures, the FOXO signaling pathway caught our attention due to its involvement in resistance to PI3K therapies [42,43] (Figure 2A). To determine whether isiPI3K treatment activated FOXO signaling in our cell line models, we monitored FOXO1 and FOXO3a localization in cells before and 24 h post-treatment with isiPI3K. Both BYL719 and GDC0032 treatments induced nuclear translocation of FOXO1 and FOXO3a (Figure 2B), supporting our conclusion that isiPI3K activates the FOXO pathway in these cells.

Among the most significantly upregulated genes in the resistant cells, we found insulin growth factor-2 (*IGF2*) (Figure 2C). Since *IGF2* has been shown to play an important role in cancer progression by limiting therapy efficacy (reviewed in Reference [44]) and has also been shown to be associated with FOXO3 activity in mesenchymal progenitor cells [45], we utilized the TCGA RNAseq data of HNC patients to explore whether expression of FOXO family members is associated with *IGF2* expression. Spearman coefficient and the −log_10_
*p*-value plot show the correlation between *IGF2* and three FOXO members (Figure 2D). 

Next, we validated *IGF2* upregulation in the resistant cell lines compared to sensitive cells using RT-PCR (Figure 2E). To test whether *IGF2* expression is upregulated following treatment with isiPI3K, sensitive UM-SCC47 and UT-SCC60A cells were treated for 24 h with GDC0032 or BYL71, respectively, and *IGF2* levels were measured by RT-PCR. Treatment of tumor cells with isiPI3K upregulated transcript levels of *IGF2* in both cell lines (Figure 2F).

### 3.3. Overexpression of IGF2 Limits the Vulnerability to isiPI3K

To investigate the contribution of IGF2 levels to the response of tumor cells to isiPI3K, we knocked down IGF2 in tumor cells using siRNA (knockdown efficacy in UT-SCC60A is shown in Appendix A) and measured cell proliferation in the absence and presence of isiPI3K. We found that reducing IGF2 expression had an additive anti-proliferative effect when combined with isiPI3K (Figure 3A). 

Next, we tested whether IGF2 directly reduces isiPI3K efficacy in our cell lines by adding recombinant human IGF2 protein (rIGF2) to the medium of isiPI3K-sensitive cells treated with isiPI3K, after which we measured cell viability. Indeed, treating UM-SCC47 and UT-SCC60A cells with isiPI3K induced growth arrest, and supplementation with rIGF2 was sufficient to rescue the growth arrest. Blocking IGF1 receptor (IGF1R) signaling with the IGF1R small molecule inhibitor, AEW541, prevented IGF2-induced rescue (Figure 3B). In addition, rIGF2 supplementation reduced isiPI3K-induced S-phase arrest (Figure 3C). Taken together, these data show that IGF2 inhibits isiPI3K efficacy.

### 3.4. IGF1R Inhibition Synergizes with isiPI3K to Inhibit Tumor Cells

Since blocking IGF1R prevented the rescue phenotype mediated by IGF2 supplementation, we explored whether blocking the IGF1R pathway with AEW541 could enhance the anti-tumor efficacy of isiPI3K in sensitive and resistant cell lines. To this end, we treated sensitive (Figure 4A, top) and resistant (Figure 4A, bottom) cell lines with GDC0032 or BYL719 in the presence or absence of AEW541. Measuring cell viability revealed that AEW541 enhanced the vulnerability of UM-SCC47 cells to GDC0032 and of UT-SCC60A cells to BYL719 in both isiPI3K resistant and sensitive cell lines. To determine whether the effect of combining isiPI3K with AEW541 was synergistic or additive, cells were seeded in 96-well plates and treated with different concentrations of each drug for 4 days, after which cell viability was determined using crystal violet staining. Calculation of drug interactions, using the ZIP synergy model [46], revealed synergism between BYL719 and AEW541 with an average score of ~11detected in UT-SCC60A, and a score of ~2 using of GDC0032 (500 nM) and of AEW541 (5 μM) in UM-SCC47. 3D surface plots showing regions of synergy with average ZIP synergy score, alongside with dose-response matrix are shown (Figure 4B). These data show that together isiPI3K and the IGF1R inhibitor function in a synergistic manner.

To conform our findings, we tested whether AEW541 enhances the anti-tumor activity of BYL719 in three additional HNC^HPV+^ cell lines, UM-SCC104, UT-SCC102, and UPCI-SCC90 (Figure 4C). In agreement with our previous findings, blocking IGF1R with AEW541 enhanced the efficacy of BYL719 in all the tested HNC^HPV+^ cell lines. 

### 3.5. Dual Treatment with isiPI3K and AEW541 Induced Growth Arrest in PDX^HPV+^ Ex Vivo

Since stimulating cells with IGF2/IGF1R was shown to activate multiple signaling pathways, including the PI3K and the mitogen-activated protein kinase (MAPK) pathways [47,48,49], we explored whether the MAPK pathway is involved in the favorable response to the drug combination. UM-SCC47 cells were treated with GDC0032, AEW541, and a combination of the two agents, after which the MAPK pathway activity was determined using western blot. While GDC0032 treatment inhibited the PI3K/AKT pathway, as indicated by a reduction of pAKT, and treatment with AEW541 altered the pAKT and slightly affected pERK, the combination of the two drugs resulted in further inhibition of the MAPK pathway (Figure 5A,B and Appendix A). Nevertheless, MAPK pathway inhibition was not observed in UT-SCC60A or in the resistant clones. These results show that the effect of the dual treatment on the MAPK pathway is more likely to be cell line specific (Appendix A).

We next aimed to validate the enhanced anti-tumor effect of isiPI3K and AEW541 in a pre-clinical model, using a single HPV+ patient-derived xenograft (PDX), designated PDX^HPV+^. To this end, we performed an ex-vivo efficacy study combining GDC0032 or BYL719 with AEW541 and using the tumor ex vivo assay (TEVA) [50]. The TEVA method enables us to explore the efficacy of a therapy combination rapidly and accurately, without having to sacrifice too many animals [50]. Briefly, PDX^HPV+^ (#7157) was sliced into 2 × 2 × 2 cubes and exposed to DMSO, GDC0032, or BYL719, with or without AEW541 for 24 h. For calculating therapy efficacy using the TEVA system, we stained the tissues with a cell proliferation marker (KI67), a cell death marker (TUNEL), and MAPK signaling pathway activation (pERK), after which the TEVA score was calculated as previously described [50] (see Material and Methods). Representative images of staining and the TEVA score showing the superior anti-tumor activity of the therapy combinations, compared to single-agent therapy, are shown in Figure 5C. To confirm that apoptosis was induced by the therapy combinations, we used anti-cleaved Caspase3 staining (Appendix A) and found increased staining in the tumors treated with the drug combinations. Taken together, these data show that treatment with isiPI3K in combination with IGF2 inhibition triggers apoptosis in HNC tumors.

### 3.6. Dual Treatment with isiPI3K and AEW541 Induced Tumor Growth Arrest in Mice 

Having found that the IGF1R inhibitor synergizes with isiPI3K in vitro and ex vivo, we next tested the impact of these agents in vivo using an HNC^HPV+^ cell line derived xenograft (CDX) and PDX^HPV+^. To obtain the CDX, we injected NSG mice with the sensitive UM-SCC47 tumor cell line and treated the mice with GDC0032 (10 mg/kg/d), AEW541 (30 mg/kg/d) or a combination of the two agents. Daily treatments of tumor-bearing mice with GDC0032 monotherapy delayed tumor growth compared to vehicle or to AEW541 treatment (Figure 6A,B). However, mice treated with the combination of GDC0032 and AEW541 exhibited tumor growth arrest, and only a minimal change in tumor volume was observed over the 15 days of treatment. The masses of the different tumors provided further support for the superior anti-tumor activity of the combination versus the single-agents (Figure 6C).

We next tested our model using PDX^HPV+^. We implanted 2-mm^3^ cubes of PDX^HPV+^ into NSG mice, and when tumors reached ~80 mm^3^, the mice were randomized into four treatment groups. Treatment of PDX^HPV+^-bearing mice with AEW541 (30 mg/kg/d) did not inhibit tumor growth, and treatment with BYL719 (25 mg/kg/d) induced a delay in tumor growth. However, the combination of BYL719 and AEW541 resulted in a superior anti-tumor effect versus single agents, as indicated by smaller tumor volumes and lower tumor weights compared to the single agents (Figure 6D–F). Notably, we did not detect significant toxicity for the therapy combination, as the weight of the animals remained stable and similar to that in the single agent groups (Appendix A). Taken together, these data show that the IGF1R inhibitor–isiPI3K combination exerts a synergistic effect in controlling HNC^HPV+^ tumors in vivo.

## 4. Discussion

Innate and acquired resistance is the major obstacle to treating cancer patients with isiPI3K [51,52,53]. Different molecular paths can lead to isiPI3K resistance, including sustained activation of mTORC1 [21,54,55], PDK1/SGK1 [56], and MYC [57]. In HNC^HPV−^, acquisition of resistance to isiPI3K is driven by increased activity of receptor tyrosine kinases (RTK), like EGFR [20,58], AXL [20,59,60], and HER3 [61], which signal through AKT or RAS/ERK pathways, or by activation of RTKs downstream of ERK/TSC2 [61], p85 [62], and CDK4/6 [63]. While, in HNC^HPV−^, different mechanisms of resistance to isiPI3K have been well documented by us and by others, in HNC^HPV+^, only innate resistance to isiPI3K has been reported [64]. Unlike the innate resistance to BYL719 in HNC^HPV+^ mediated by upregulating ERBB2/3 signaling [64], acquired-resistant cells did not upregulate HER3 or AXL expression (Appendix A). Here we found that a ligand of IGF1R, IGF2, is upregulated in response to isiPI3K treatment, and that IGF2 upregulation limits isiPI3K efficacy. To the best of our knowledge, this is the first report of IGF2-driven resistance to PI3K therapy in cancer.

IGF2, a well-studied growth factor, is part of the insulin/IGF-signaling axis, regulating cell proliferation, survival, and metabolism (reviewed in Reference [49]). IGF2 binds to the IGF1R and activates survival pathways, including AKT/mTOR and MAPK [65]. Upregulation of IGF2 in solid tumors is well documented in a variety of cancers, including colorectal [66], ovarian [67], esophageal [68] cancers, and in acute myeloid leukemia [69], and TCGA analysis shows that IGF2 is also overexpressed in HNC versus normal tissue (Appendix A). It has been shown that upregulation of IGF2 drives chemoresistance to Taxol (paclitaxel), fluorouracil and CDDP, in ovarian cancer [70,71], esophageal cancer [72], and HNC [73], respectively. In addition, it has also been shown that overexpression of IGF2 induces resistance to targeting agents, such as the EGFR inhibitors, osimertinib and erlotinib, by re-activating the AKT and MAPK pathways in lung cancer and cholangiocarcinoma [74]. Here, we found that reducing IGF2 expression levels or blocking IGF1R, using AEW541, combined with either isiPI3K, BYL719 or GDC0032, enhanced tumor growth arrest. In a previous study, we found that IGF1 activates mTOR and limits the antitumor activity of BYL719 in breast and ovarian cancers [21,75]. In agreement with our findings, Fox et al. showed that treatment with the dual IGF-IR/InsR inhibitor, AZD9362, enhanced the anti-tumor effect of the AKT inhibitor, AZD5363, in ER-positive breast cancer cells [76], indicating that the IGF axis plays a role in maintaining survival signaling pathways and limiting the efficacy of PI3K and AKT inhibitors.

In our previous study, up-regulation of IGF2 was not observed in HNC^HPV−^ BYL719-acquired resistance cell line models, and supplementation of tumor cells with recombinant IGF was able to achieve a partial rescue to anti-PI3K in two of the five tested cell lines [20]. Therefore, it is reasonable to conclude that IGF2 upregulation plays a greater role in acquired resistance to isiPI3K in HNC^HPV+^ than in HNC^HPV−^. The molecular explanation behind the difference in responses of HNC^HPV+^ and HNC^HPV−^ cells to isiPI3K is still unclear. One possible explanation for this phenomenon is the difference in p53 function, as in HPV^+^ cells p53 is genetically intact, while, in HPV− cells, p53 is either deleted or mutated [18]. It is known that IGF2 expression is required for the viability of p53 null mice and that IGF2 promotes tumor initiation upon expression in p53 heterozygous mice [77], supporting the model in which p53 is involved in the cellular response to IGF2. Another explanation may be related to the p53 transcriptional activity in HNC^HPV+^ cells, as p53 was shown to repress IGF2 transcription [78], and PI3K therapy inhibited p53 [79]. An additional explanation may be related to the inactivation of p53 by HDAC8, whose expression is induced by FOXO1 and FOXO3a [80].

To date, agents that block the PI3K pathway have shown modest effects as monotherapy in *PIK3CA* amplified/mutated solid tumor patients, including in HNC [27,32,33,35,81,82]. Therefore, it is important to identify compounds that synergize with isiPI3K for treating cancer patients [58,59,75,83,84,85,86]. Indeed, the FDA recently approved BYL719 (alphelisib) for treating *PIK3CA*-mutated breast cancer patients in combination with the estrogen receptor degrader, fulvestrant, following a randomized phase 3 clinical trial that showed that a combination of the two agents resulted in improvement in patient outcomes [31]. The success of isiPI3K in combination with another targeting agent highlights the need to further develop such inhibitors for cancer patients [87]. Currently, isiPI3Ks, such as BYL719, GDC0032, AZD8186, and GSK2636771, are under clinical development for treating lymphoma and solid cancers (Appendix A). According to clinicaltrials.gov, at present (2021), there are 18 ongoing phase 2 and phase 3 clinical trials testing therapy combinations with multiple agents targeting the estrogen receptor (fulvestrant and letrozole), the HER2 receptor (trastuzumab and pertuzumab), the androgen receptor (enzalutamide), cyclin-dependent kinases CDK4/6 (LEE011), and the PD-1 receptor (pembrolizumab), in addition to chemotherapy (paclitaxel and nab-paclitaxel) and radiotherapy. This large number of trials highlights the apparent importance of the PI3K pathway in tumor progression and the role of drug combinations in enhancing the efficacy of anti-PI3K therapies. Our findings show that the combination of isiPI3K and IGF1R inhibition is potent in preclinical models of HNC^HPV+^; nevertheless, the contribution of the *PIK3CA* status (mutated/amplified/WT) to the combination therapy response is yet to be determined.

Despite the importance of IGF1R as an anti-cancer target, the clinical benefits of IGF1R inhibitors are still limited mainly due to hyperinsulinemia and hyperglycemia toxicity (reviewed in Reference [88]). A phase Ib/II clinical trial testing the combination of BYL719 and an anti-IGF1R antibody, AMG479, showed toxicity in cancer patients (NCT 01708161), but there are also other approaches to targeting the IGF2/IGF1R axis. Specifically, IGF-1/2 neutralizing antibodies that block IGF-induced IGF-1R and INSR-A functions, xentuzumab and Dusigitumab, have been tested in preclinical models [89,90,91,92]. Currently, there are five ongoing clinical trials testing the therapy efficacy of xentuzumab (BI 836845) in combination with enzalutamide or anti-hormone therapies, in breast cancer (NCT02123823, NCT03659136), prostate cancer (NCT02204072) and neoplasms (NCT02145741, NCT03099174). Another alternative approach to targeting the IGF axis is via IGF ligand TRAPs [93,94,95]. Although there is currently no clinical use of IGF ligand TRAPs, preclinical data seem to be promising [95].

In summary, our findings highlight the likelihood that the factors governing the acquisition of resistance in HNC^HPV+^ may be distinct from those in HNC^HPV−^. While, in HNC^HPV−^, expression of RTKs increases following PI3K treatment [20,58,59,60,61], in HNC^HPV+^, tumor cells acquire resistance to isiPI3K by the secretion of IGF2 to activate IGF1R. Our results provide the first evidence that blocking the IGF2/IGF1R pathway in combination with isiPI3K is a promising therapeutic strategy for treating HNC^HPV+^ patients.

## 5. Conclusions

IGF2 expression increases in HNC^HPV+^ following treatment with BYL719 and GDC0032 (isiPI3K). IGF2 limits the vulnerability of HNC^HPV+^ to isiPI3K via the activation of the IGF1R. Co-targeting of PI3K and IGF1R are potent in HNC^HPV+^ models. These findings provide the motivation for testing the combination of PI3K and IGF1R inhibitors in HNC^HPV+^ patients.

## Figures and Tables

**Figure 1 cancers-13-02250-f001:**
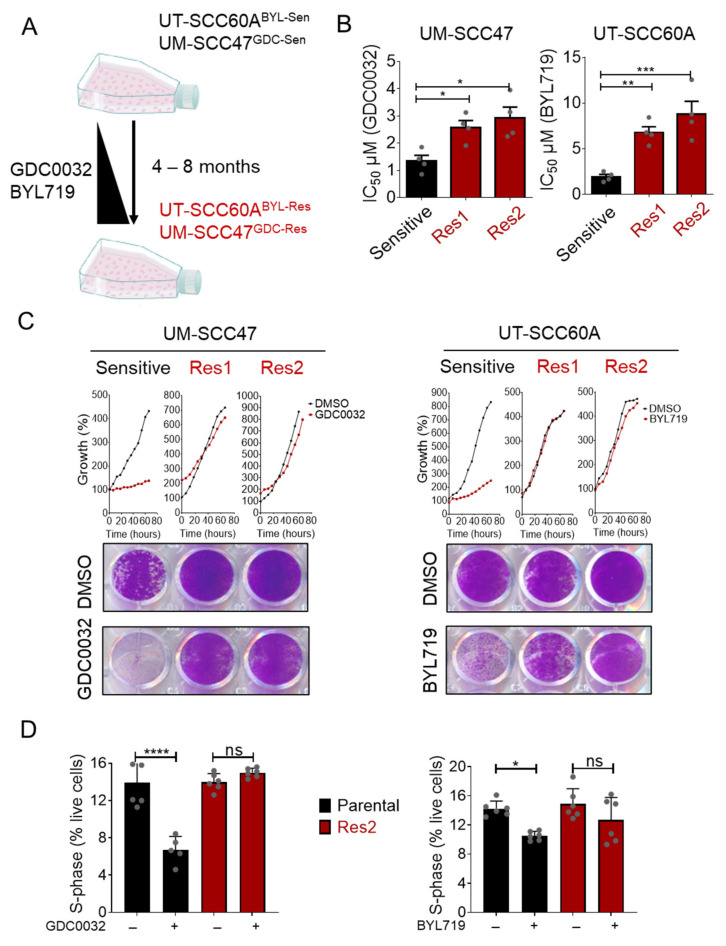
Generation of isiPI3K-acquired resistance HNC^HPV+^ cell lines. (**A**) The indicated cell lines were cultured with increasing concentrations of either GDC0032 (100 nM up to 1 μM) or BYL719 (1 μM up to 4 μM) for 4 to 8 months, until resistance emerged. (**B**) One sensitive and two resistant cell lines (Res1 and Res2) of UM-SCC47 and of UT-SCC60A were cultured with DMSO and increasing concentrations of GDC0032 (0–2 μM) or BYL719 (0–10 μM) for 4 days. Cell viability was determined using crystal violet staining, and IC_50_ values were calculated. IC_50_ results of 4 separate experiments are presented as means ± SEM. Biological replicates from separate experiments are shown as grey dots. (**C**) UM- SCC47 and UT-SCC60A sensitive and resistant cell lines (Res 1 and Res2) were cultured with DMSO, GDC0032 (500 nM), or BYL719 (2 μM) for 4 days. A live cell imager monitored cell growth every 6 h, and cell confluency was calculated. At the end of the experiment, cells were stained with crystal violet for determination of final cell densities. Each proliferation experiment was repeated 3 separate times, and a representative experiment is presented. (**D**) UM- SCC47 and UT-SCC60A sensitive and resistant cell lines (Res2) were cultured with DMSO, GDC0032 (500 nM), or BYL719 (2 μM). Three days post-treatment, cell cycle analysis was performed, and the fraction of the cells in S-phase was determined. Cell cycle results of 2 separate experiments are presented as means ± SEM. Biological replicates from separate experiments are shown as grey dots. Statistical significance was calculated by one-way ANOVA, * *p* < 0.05; ** *p* < 0.01; *** *p* < 0.001; **** *p* < 0.0001.

**Figure 2 cancers-13-02250-f002:**
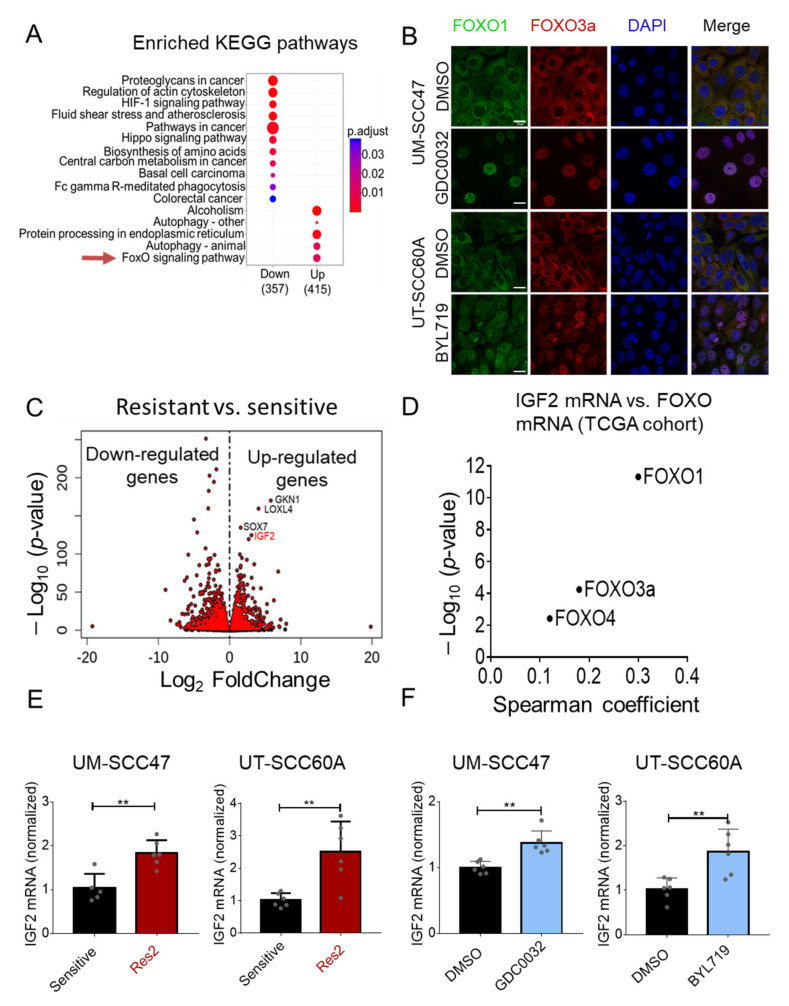
isiPI3K resistant cells activate FOXO signaling pathway and upregulate the expression of IGF2. (**A**) The transcriptome of sensitive and resistant (Res2) UM-SCC47 and UT-SCC60A cells was profiled using RNAseq in 3 biological replicates per each cell line. Up- or down-regulated genes of UM-SCC47 and UT-SCC60A (resistant vs. sensitive) cells were identified using an adjusted *p*-value < 0.05 & FC > 1 as a cutoff. KEGG pathway enrichment analysis was performed on the list of genes whose expression was significantly different between the resistant and non-resistant cells. (**B**) UM-SCC47 and UT-SCC60A were cultured with DMSO, GDC0032 (500 nM), or BYL719 (2 μM) for 24 h, after which cells were fixed and stained using the indicated antibodies. The immunofluorescence experiment was repeated 3 times, and representative stained images of FOXO1, FOXO3a and DAPI from one of the experiments are presented. Scale bar: 20 μm. (**C**) Volcano plot visualizing the dysregulated genes (log2 fold change against −log10 *p*-value) between sensitive and resistant cells. (**D**) TCGA (Firehose Legacy) data show the correlation (Spearman coefficient against −log10 *p*-value) between *IGF2* mRNA and mRNA levels of FOXO family members in HNC cancer patients. (**E**) RT-PCR analysis measuring *IGF2* mRNA levels in sensitive UM-SCC47 and UT-SCC60A and resistant cells. (**F**) RT-PCR analysis determining *IGF2* mRNA levels in UM-SCC47 and UT-SCC60A cell lines following 24 h of treatment with DMSO, GDC0032 (0.5 μM), or BYL719 (2 μM). RT-PCR results of 2 separate experiments are presented as means ± SEM. Biological replicates from separate experiments are shown as grey dots. Statistical significance was calculated by an unpaired *t*-test. ** *p* < 0.01.

**Figure 3 cancers-13-02250-f003:**
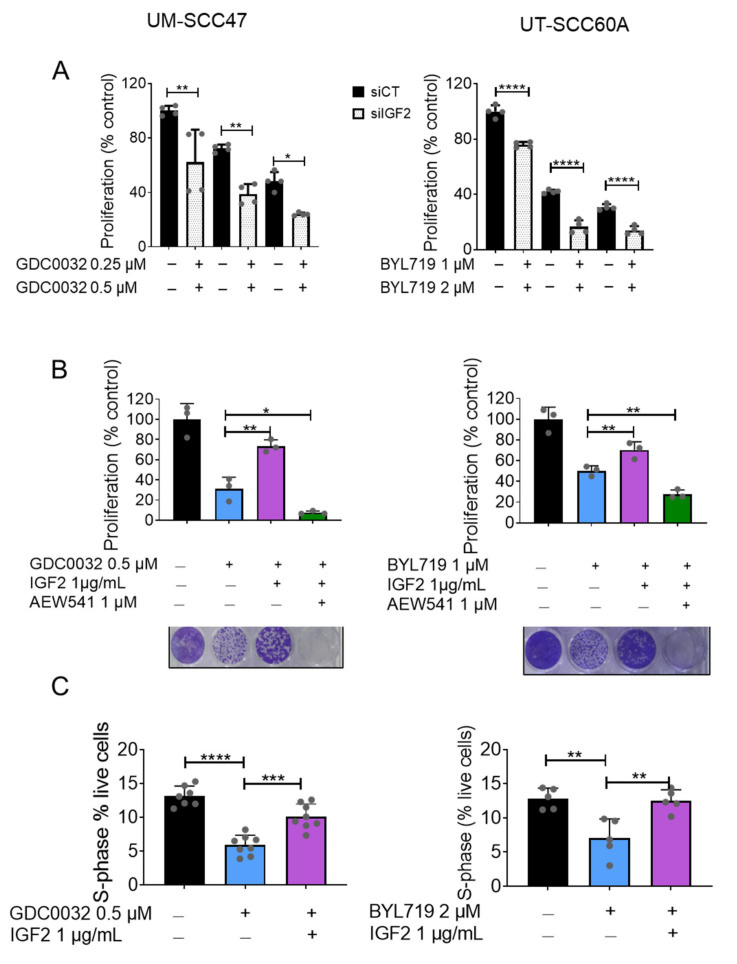
IGF2 limits the response of HNC^HPV+^ cells to isiPI3K. (**A**) UM-SCC47 and UT-SCC60A sensitive cells were transfected with either si-Control (siCT) or si-RNA targeting IGF2 (siIGF2) in the presence of DMSO, GDC0032, or BYL719. Cell viability was determined 4 days post-treatment using crystal violet staining. siRNA results of 2 independent experiments are presented as means ± SEM. Biological replicates from separate experiments are shown as grey dots. (**B**) UM-SCC47 and UT-SCC60A sensitive cells were cultured with DMSO, GDC0032, or BYL719, and AEW541, in the presence or absence of rIGF2. Cell viability was measured 4 days post-treatment using crystal violet staining. The proliferation experiment was repeated 3 times, and the results of a representative experiment are presented as means ± SEM. Biological replicates from one experiment are shown as grey dots. (**C**) UM- SCC47 and UT-SCC60A sensitive cells were cultured with DMSO, GDC0032 (500 nM), or BYL719 (2 μM), with or without rIGF2. Three days post-treatment, cell cycle analysis was performed. The fraction of cells in the S-phase was determined. Cell cycle results of 2–3 separate experiments are presented as means ± SEM. Biological replicates from separate experiments are shown as grey dots. Statistical significance was calculated by one-way ANOVA, * *p* < 0.05; ** *p* < 0.01; *** *p* < 0.001; **** *p* < 0.0001.

**Figure 4 cancers-13-02250-f004:**
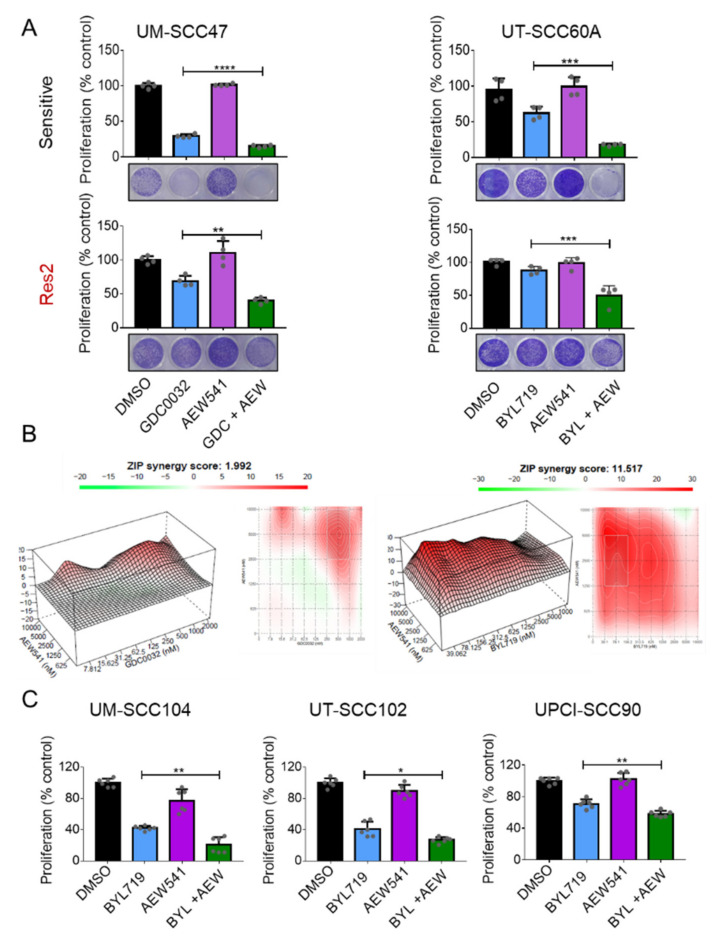
IGF1R inhibition synergizes with isiPI3K in inhibiting the viability of HNC^HPV+^ cells. (**A**) UM-SCC47 and UT-SCC60A sensitive and acquired resistance cells (Res2) were cultured with DMSO, GDC0032 (500 nM) or BYL719 (2 μM), AEW541 (1 μM), or a combination of AEW541 with GDC0032 or BYL719. Four days post-treatment, cell viability was determined using crystal violet staining. The proliferation experiment was repeated 3 times, and representative results from one experiment are presented. Biological replicates from one experiment are shown as grey dots. (**B**) ZIP model synergy test was performed in both UM-SCC47 and UT-SCC60A cell lines. Cells were cultured with increasing concentrations of GDC0032 (0 up to 2 μM) or BYL719 (0 up to 10 μM) and AEW541 (0 up to 10 μM). Four days post-treatment cell viability was determined using crystal violet stanning. The results are presented as a ZIP synergy score with 3-D surface plots displaying synergy regions (left panel) along with a dose-response matrix (right panel). The synergy experiment was repeated 3 times, and representative results from one experiment are presented. (**C**) UM-SCC104, UT-SCC102, and UPCI-SCC90 sensitive cells were cultured with DMSO, BYL719 (2 μM), AEW541 (1 μM), or a combination of the two drugs. Four days post-treatment cell viability was determined using crystal violet staining. The proliferation experiment was repeated 3 times, and the results of one representative experiment are presented as means ± SEM. Biological replicates from one experiment are shown as grey dots. Statistical significance was calculated by one-way ANOVA, * *p* < 0.05; ** *p* < 0.01; *** *p* < 0.001; **** *p* < 0.0001.

**Figure 5 cancers-13-02250-f005:**
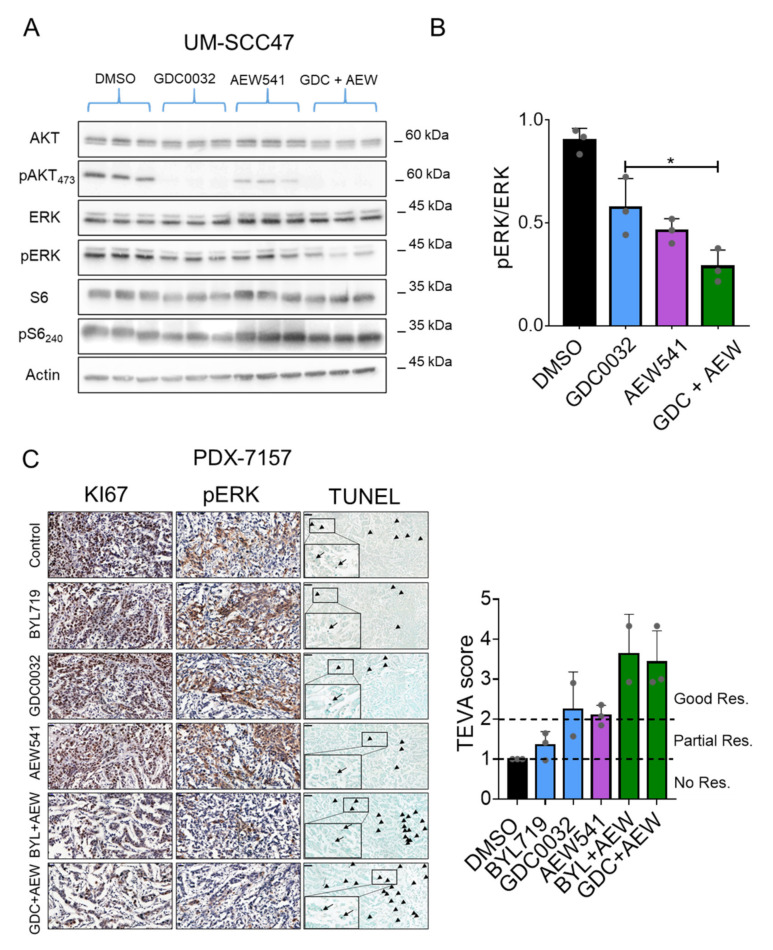
isiPI3K and AEW541 induced growth in a PDX^HPV+^ ex vivo model. (**A**) UM-SCC47 sensitive cells were treated for 4 h with DMSO, GDC0032 (500 nM), AEW541 (1 μM), or a combination of the two drugs, after which cells were lysed. The indicated proteins and phosphorylated proteins were measured using western blot. (**B**) pERK plot was quantified with Adobe Photoshop 2020and normalized to ERK. The western blotting experiment was repeated 3 times, and representative results from one experiment are presented. (**C**) PDX^HPV+^ was cultured with DMSO, GDC0032 (500 nM) or BYL719 (2 μM), AEW541 (1 μM), or a combination of AEW541 with GDC0032 or BYL719, as indicated, for 24 h, after which the tissue was fixed and analyzed using IHC. Representative images of IHC staining for KI67, pERK, and TUNEL are presented together with the TEVA score. Scale bar for KI67 and MAPK: 20 μm, for TUNEL: 50 μm. Statistical significance was calculated by one-way ANOVA, * *p* < 0.05.

**Figure 6 cancers-13-02250-f006:**
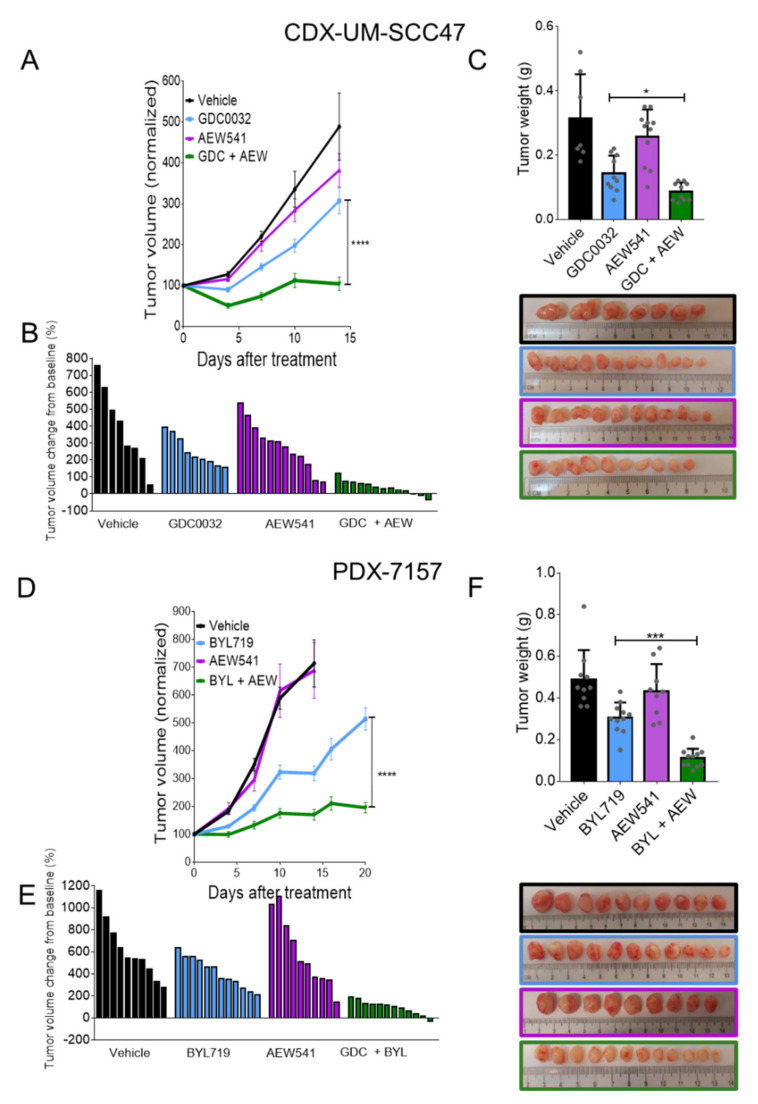
Dual treatment with isiPI3K and AEW541 arrested tumor growth in mice. UM-SCC47 and PDX ^HPV+^ were injected or implanted subcutaneously (s.c.) into NSG mice. After tumors developed (tumor volume ~100 mm^3^), mice were randomized into 4 groups (tumors n = 8–12) and treated daily with the treatment indicated below. (**A**) Growth kinetics of UM-SCC47 tumors following treatment via oral gavage with vehicle, GDC0032 (10 mg/kg), AEW541 (30 mg/kg) or a combination of the two drugs. (**B**) Waterfall plot showing the change in the volume of UM-SCC47 tumors on endpoint day (15 days after starting the treatment) compared to the baseline for all treatment groups. (**C**) Tumor weight and images of UM-SCC47 CDX on endpoint day (15 days after starting the treatment). (**D**) Tumor growth kinetics of PDX ^HPV+^ following treatment via oral gavage with vehicle, BYL719 (25 mg/kg), AEW541 (30 mg/kg), or a combination of the two drugs. Tumor growth was monitored using a caliper every 3 days. (**E**) Waterfall plot showing the change in tumor volume on the endpoint day (15–20 days after starting the treatment) compared to the baseline for all treatment groups. (**F**) Tumor weight and images of PDX ^HPV+^ on the endpoint day (15–20 days after starting the treatment). Tumor volumes were normalized to initial volumes and presented as averaged percentages of the initial volumes ± SEM. Statistical significance was calculated using a two-way ANOVA. * *p* < 0.05; *** *p* < 0.001; **** *p* < 0.0001.

## Data Availability

The data presented in this study are available in this article (and Appendix A).

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
