# Peer review of "IGF2 Mediates Resistance to Isoform-Selective-Inhibitors of the PI3K in HPV Positive Head and Neck Cancer"

_cancers, 2021, doi:10.3390/cancers13092250_

Round 1
Reviewer 1 Report
This is an interesting and elegant study delineating the molecular mechanisms at the basis of resistance to two inhibitors (isiPI3K) alpelisib (BYL719) and Taselisib 25 (GDC0032) in human papillomavirus positive head and neck cancer. The authors employ RNA sequencing of isiPI3K-sensitive tumour cells and their corresponding isiPI3K-acquired-resistant tumour cells and identify IGF2 critically associated to drug resistance.
The study is well-performed and highly relevant for Cancers. The article is also well-written and the data well-presented. There is only a marginal number of minor points to address:
- From the TUNEL assay it looks like that apoptosis is involved as modality of cell death associated to the mechanism of resistance. This would imply that IGF2 has an impact on the ability of the cell to undergo apoptosis. Can the authors test caspases activation as a formal proof that IGF2 is regulating apoptosis in this mechanism of drug resistance?
- It would be interesting to assess the impact of IGF2/IGFR expression on a cohort of HPV+ HNSCC patients, in term of drug response and prognosis. Can the authors test this by querying gene expression datasets?
- Do the authors predict that this mechanism is specific of HPV+ HNSCC or also conserved in HPV-? The authors should comment this in the discussion and/or test it by bioinformatic on gene expression patients analyses.
- Please make sure to include scale bars to the microscopy figures.
Author Response
We are grateful for your encouraging and thoughtful comments and suggestions regarding our
original submission. In response to these comments, we have made a number of modifications to
our manuscript and have performed additional experiments. Below we detail the Reviewers’
comments in black and our responses and modifications in blue.

Reviewer 2 Report
Isoform-specific inhibitors of p110alpha PI3K hold promise in treating tumors with PIK3CA mutations, but their efficacy is limited by intrinsic and acquired resistance, and likely necessitates combination therapies. Badarni et al. investigate the mechanisms of resistance to isiPI3K in PIK3CA-mutant HPV+ head and neck cancer cell lines. They generated cell line models of acquired resistance to alpelisib and taselisib by exposing cells to increasing drug concentrations until resistance developed, then profiled resistant vs sensitive cells by RNA-seq. They found that IGF2 was upregulated in both models of resistant cells and showed that IGF2 upregulation was causal to resistance, which could be blocked by an IGF1R inhibitor, and the combination of isiPI3K and an IGF1R inhibitor increased efficacy in vivo. Overall, their conclusions are sound, but the manuscript would benefit from more in-depth characterization of the resistant cell lines and a clearer description of experimental methods. Also, limitations of the proposed clinical strategy of combining a PI3K inhibitor and IGF1R inhibitor should be discussed.
Major points:
- Methods/figure legends require clearer descriptions. For the RNA-seq, were both samples drug-treated, or was RNA extracted in the absence of drug? If the latter, how long were the resistant cells cultured without drug before extraction? How many biological or technical replicates per condition?
- Figure 1: further characterization of the resistant cells would strengthen the manuscript. Are the BYL-resistant cells cross-resistant to GDC, and vice-versa? Are they resistant to Pan-PI3K inhibitors, or the combination of p110alpha + p110beta inhibitors? Is the resistance stable, ie, if the cells are cultured in the absence of drug for >2 weeks, do they maintain resistance? A western blot should also be shown, in untreated vs treated sensitive vs resistant cells: does BYL/GDC treatment still block P-AKT in resistant cells? What about downstream targets, like P-S6?
- Figure 2: Again, this is difficult to interpret without knowing whether RNA was extracted from samples in the presence of drug. Are IGF-regulated genes upregulated in resistant cells, or an IGF pathway gene signature? Is this simply because the resistant cells were treated with drug and the sensitive cells were not, because short-term drug treatment increases IGF2 expression? (therefore, this would be more of an adaptive rather than acquired mechanism). Is P-IGF1R increased in resistant cells?
- Figure 3: knockdown efficiency of IGF2 siRNA should be shown, and ideally a western blot showing decreased P-IGF1R.
- Figure 3B: the effect of AEW541 alone should be shown.
- Figure 5A: multiple cell lines, and ideally, resistant cell lines should be shown, to see if the effect of the combination on P-ERK is generalizable.
- Previous studies have implicated IGF1 and the IGF1R pathway in compensation to PI3K/AKT pathway blockade in breast cancer, including from the senior author. Also, AKT inhibition has been shown to result in upregulation of IGF2 expression. The authors should cite these studies in the discussion. PMID: 23844554; PMID: 23903756
- Clinical studies of alpelisib + IGF1R inhibitors have shown limited efficacy and high toxicity, which could limit the clinical relevance of this study. a phase Ib trial of alpelisib with the IGF1R mAb ganitumab (AMG479; clinicaltrials.gov NCT 01708161) resulted in excessive rash and hyperglycemia and limited clinical activity in PIK3CA-mutated or amplified solid tumors, resulting in termination of the study (https://clinicaltrials.gov/ct2/show/results/NCT01708161?term=NCT01708161&rank=1). In the Discussion, the authors should discuss this limitation and suggest alternative approaches that might be more tolerable/enhance efficacy. Also, did the combination of PI3Ki and IGF1R inhibitor affect the weight of the mice, or did the mice show other signs of toxicity?
- In the discussion, the authors suggest that resistance to PI3Ki in HPV- HNCs is due to RTK upregulation, and resistance in HPV+ HNCs is due to IGF2 upregulation. Did the authors rule out upregulation of RTK expression in their HPV- resistant cells?
Minor points:
- I’m not sure if taselisib can truly be called isoform-specific, since it blocks 3 out of 4 PI3K isoforms.
- Figure 1B: a representative dose-response curve should be shown in the supplemental data.
- “TEVA score” requires additional definition. Is this a composite of p-ERK, TUNEL, and Ki67?
- Supplementary Fig 1: show p-value
- There are several typos throughout the manuscript:
- Line 95: “spaing” should be “sparing”
- Line302: “sperate” shoule be “separate”
- Figure 2C: “resistance” should be “resistant”
- Line 501: “AOVA” should be “ANOVA”
- Line 564: “RTKs expression” should be “expression of RTKs”
- Line 575: “gen” should be “gene”
- Line 575: “trails refernce” should be “trials reference”
- There are likely others. Please check.
Author Response

(The authors gave the same response as above.)

Round 2
Reviewer 2 Report
The authors have sufficiently addressed my concerns. I would recommend including the western blots showing no upregulation of HER3/AXL in the resistant cells as supplementary data, to further differentiate the mechanisms of resistance between HPV- and HPV+ HNC cells.
Author Response
We appreciate the reviewers' comment. We have added the WB data to the supplementary Figures and we have modified the text in line 543.